# Serum Phospholipids Are Potential Therapeutic Targets of Aqueous Extracts of Roselle (*Hibiscus sabdariffa*) against Obesity and Insulin Resistance

**DOI:** 10.3390/ijerph192416538

**Published:** 2022-12-09

**Authors:** Carmen Alejandra Rangel-García, Rosalía Reynoso-Camacho, Iza F. Pérez-Ramírez, Elizabeth Morales-Luna, Ericka A. de los Ríos, Luis M. Salgado

**Affiliations:** 1Centro de Investigación en Ciencia Aplicada y Tecnología Avanzada del Instituto Politécnico Nacional, Colinas del Cimatario, Queretaro 76090, Qro., Mexico; 2Facultad de Química, Universidad Autónoma de Querétaro, C.U. Cerro de las Campanas, Queretaro 76010, Qro., Mexico; 3Instituto de Neurobiología, Universidad Nacional Autónoma de México, Campus Juriquilla, Juriquilla, Queretaro 76230, Qro., Mexico

**Keywords:** *Hibiscus sabdariffa*, in vitro study, obesity, insulin resistance, phospholipids, targeted metabolomics

## Abstract

Roselle (*Hibiscus sabdariffa*) is rich in phenolic compounds with antiobesogenic and antidiabetic effects. In this study, the effects of aqueous extracts of two varieties of *Hibiscus sabdariffa*, Alma blanca (white-yellow color) and Cuarenteña (purple color), were evaluated for the prevention of obesity and insulin resistance in rats fed a high-fat and high-fructose diet (HFFD), identifying targeted molecules through global metabolomics. After sixteen weeks, both roselle aqueous extracts prevented body weight gain, and white roselle extract ameliorated insulin resistance and decreased serum free fatty acid levels. Moreover, white roselle extract decreased 18:0 and 20:4 lysophosphatidylethanolamines and purple roselle extract increased 16:0 and 20:4 lysophosphatidylinositol compared to HFFD-fed rats. These results demonstrate that roselle’s beneficial health effects are variety-dependent. Interestingly, the white roselle extract showed a greater beneficial effect, probably due to its high contents of organic and phenolic acids, though its consumption is not as popular as that of the red/purple varieties.

## 1. Introduction

Obesity is an excessive fat accumulation that leads to insulin resistance (IR) and type 2 diabetes mellitus (T2DM). This IR is related to adipose tissue lipolysis and the release of free fatty acids (FFA) into the blood. This lipolysis is a positive regulator of hepatic gluconeogenesis and an inhibitor of glucose uptake in muscle and adipose tissue. Thus, free fatty acids are the fasting hyperglycemia inductors that define T2DM [1]. Therefore, insulin resistance is related to lipid metabolism disorders. Specifically, serum phospholipids have been identified as metabolites involved in obesity and IR development [2]. Hence, these molecules could be a relevant target for novel treatments for these metabolic disorders.

Dietary strategies used to prevent the development of obesity and IR include the consumption of plant extracts rich in phenolic compounds with pharmacological activity, such as roselle (*Hibiscus sabdariffa)* calyx (HSC) aqueous extracts [3]. Many in vitro and in vivo studies have demonstrated the antiobesogenic, antidiabetic, and anti-hyperlipidemic effects of red varieties of HSC extracts [4,5,6]. The mechanisms proposed for these effects include the inhibition of fatty acid synthesis [7], improved beta-oxidation, ameliorated systemic inflammation [8], the inhibition of digestive enzyme activity, a reduction in oxidative stress [9], increased blood glucose uptake by adipocytes [10], and the improvement of endothelial dysfunction [11].

However, there are many varieties of *Hibiscus sabdariffa*, which provide a wide range of colored calyces, from light yellow to pink, red, and deep purple colors. The light yellow color is shown by variety known as Alma blanca, which comes from a natural mutation of a Criolla red variety [12]. A contrasting variety is the Cuarenteña, which produces purple calyces. In a study in which 26 varieties produced in Mexico were collected, these two varieties showed high yields of 570 kg/ha and 512.4 kg/ha of calyces, respectively; both grow in tropical climates [13]. 

The color of calyces is associated with the phytochemical content. For instance, purple calyces have greater anthocyanin concentrations than red and pink calyces, whereas white varieties show no contents of these pigments [14]. Interestingly, despite these differences, we previously demonstrated that the Alma blanca variety (light yellow calyx color) exerts a slightly greater preventive effect on body weight gain and adipocyte hypertrophy than the Criolla variety (red calyx color). These effects were mainly associated with the high contents of organic acids (hibiscus acid, hydroxycitric acid, and dimethyl hibiscus acid) identified in Alma blanca variety [15].

These results demonstrated that the beneficial effect of HSC aqueous extracts is not only related to roselle anthocyanins. Therefore, it is necessary to evaluate the beneficial health effects of HSC aqueous extracts of varieties with different phytochemical compound profiles. In this regard, this study aimed to evaluate the preventive effects of the consumption of Alma blanca (light yellow color) and Cuarenteña (purple color) HSC aqueous extracts on obesity and insulin resistance in rats fed a high-fat and high-fructose diet (HFFD) and to identify their molecular targets through global targeted metabolomics. 

## 2. Materials and Methods

### 2.1. Reagents 

Formic acid, Folin–Ciocalteu reagent, ABTS (2,2′-azino-bis(3-ethylbenzo-thiazoline-6-sulfonic acid), DPPH (2,2-diphenyl-1-picrylhydrazyl), 6-hydroxy-2,5,7,8-tetramethylchroman-2-carboxylic acid (Trolox), quercetin rutinoside, myricetin, quercetin, vanillic acid, protocatechuic acid, chlorogenic acid, caffeic acid, coumaric acid, ferulic acid, sinapic acid, and citric acid were obtained from Sigma–Aldrich Chemical Company (St. Louis, MO, USA). Sodium carbonate, sodium nitrite, aluminum chloride, potassium chloride, sodium acetate, and solvents were purchased from J. T. Baker (Phillipsburg, NJ, USA). All reagents used for the preparation of polyphenols and spectrophotometric determinations were of analytical grade. Water and acetonitrile were mass grades and were acquired from Thermo Fisher Scientific Inc. (Waltham, MA, USA). 

### 2.2. Herbal Materials and Aqueous Extract Preparation 

*Hibiscus sabdariffa* calyces from Alma blanca (white-yellow color) and Cuarenteña (purple color) varieties were obtained from farmers from Oaxaca, México, and Nayarit, Mexico, respectively. The aqueous roselle extracts were prepared following the popular procedure for elaborating a roselle drink in Mexico. The aqueous extracts were prepared at 3% dry calyces through a decoction process. Briefly, sun-dried calyces were boiled for 15 min, cooled at room temperature, filtered, and stored at 4 °C protected from light for further analysis [16].

### 2.3. Quantification of Total Polyphenols

#### 2.3.1. Quantification of Total Phenolic Compounds

The total phenolic compounds were quantified according to Singleton et al. [17]. First, 200 µL of a sample and 800 µL of distilled water were diluted. A 10 µL aliquot of this solution was mixed with 40 µL of distilled water, 25 µL of the Folin–Cicalteu 1 N reagent, and 125 µL of Na_2_CO_3_ (20%). After that, the mixture was incubated for 30 min, and the absorbances were measured at 765 nm. The total phenolic compound quantification was performed using a standard gallic acid curve, and the results were expressed as mg of gallic acid equivalents/mL of beverage. 

#### 2.3.2. Quantification of Total Flavonoids

The total flavonoids were quantified using a colorimetric assay method [18]. An aliquot of 0.25 mL of the extract was mixed with 75 µL of a 5% NaNO_2_ solution, 0.150 mL of a 10% AlCl_3_ solution, and 0.5 mL of a 1 M NaOH solution. The final volume was adjusted to 2.5 mL with distilled water. The mixture was kept at room temperature for 5 min, and the absorbance was measured at 510 nm. A calibration curve was recorded using (+) catechin, and the total flavonoid content was expressed as μg of (+) catechin equivalents/mL of beverage.

#### 2.3.3. Quantification of Monomeric Anthocyanins

The total monomeric anthocyanin content was determined by the pH differential method [19]. First, samples (50 µL) were diluted with 175 µL of each buffer solution (0.025 M potassium chloride, pH 1, and 0.4 M sodium acetate, pH 4.5). After that, the absorbances were measured at 510 nm and 700 nm, considering both pHs. The results were expressed as mg of cyanidin-3-O-glucoside/mL of beverage.

### 2.4. Polyphenol and Organic Acid Profile

HSC aqueous extracts were filtered (Waters Co, Milford, MA, USA, Acrodisc Syringe Filter, PVDF, 25 mm, 0.45 μm) and injected (6 μL) into a BEH Acquity C18 column (2.1 × 100 mm × 1.7 μm, 35 °C) installed in an Ultra-Performance Liquid Chromatograph (UPLC) coupled to a diode array detector (DAD) and an electrospray ionization (ESI)-quadrupole/time-of-flight mass spectrometer (MS) (ESI-QToF MS, Vion IMS, Waters Co., Milford, MA, USA). 

The chromatographic separation was achieved with the following mobile phases: (A) water with 0.1% formic acid (*v/v*) and (B) acetonitrile at a flow rate of 0.5 mL/min under gradient conditions: 0% B/0 min, 15% B/2.5 min, 21% B/10 min, 90% B/12 min, 95% B/13 min, 0% B/15 min, and 0% B/17 min. The MS conditions were as follows: 2.0 kV of capillary voltage, 40 eV of cone voltage, 6 V for low collision energy, 15–45 V for high collision energy, a 120 °C source temperature, 50 L/h of N_2_ as cone gas flow, and 800 L/h at 450 °C of N_2_ as a desolvation gas. 

Data acquisition was carried out in the -ESI (organic acids, flavones, flavonols, and phenolic acids) and +ESI (anthocyanins) ionization modes within a 100–1200 Da mass range. A leucine–enkephalin solution (50 pg/mL) was used for lock mass correction at 10 mL/min. Identification was carried out by analyzing the exact mass of the pseudomolecular ion (mass error < 10 ppm) and fragmentation pattern using the UNIFI software (Waters Co) [20]. Calibration curves were constructed with a commercial standard for quantification. The results were expressed as μg/mL of HSC aqueous extracts. The polyphenol and organic acid profile was determined using three independent replicates of the HSC aqueous extracts.

### 2.5. Free Radical Scavenging Assays

The antioxidant capacities of the extracts were measured using the DPPH radical (2,2-diphenyl-1-picrylhydrazyl, DPPH•), according to the procedure of Brand-Wiliams et al. [21]. The extract (0.2 mL) was mixed with 0.8 mL of 0.3 mM DPPH in ethanol. The mixture was shaken and left at room temperature for 10 min in the dark. Finally, the change in the absorbance was measured at 520 nm.

ABTS (2,2’-azino-bis-3-ethylbenzo-thiazolin-6-sulfonic acid) radical scavenging was carried out as described by Re et al. [22]. The ABTS^•+^ radical was generated by reacting a 2.45 mM potassium persulfate solution with a 7 mM aqueous ABTS solution. The mixture was kept in the dark at room temperature for 16 h before use. The ABTS^•+^ solution was diluted with methanol to an absorbance of 0.70 ± 0.02 at 730 nm. Then, 0.2 mL of the extracts were mixed with 3.9 mL of the diluted ABTS^•+^ solution, and the absorbance reading was taken every 20 s for 6 min. 

The free radical (DPPH and ABTS) scavenging capacities were expressed as μmol ET/100 mL.

### 2.6. Animals and Treatments

Thirty-two male Wistar rats weighing 160 to 210 g were acquired at the Universidad Nacional Autónoma de México (Querétaro, Mexico) and were housed in standard environment conditions (40–60% relative humidity, 24 ± 2 °C, 12 h light/12 h dark cycle) and fed commercial rat food (Rodent Lab Chow 5001, Purina Nutriments) and tap water ad libitum. The animal procedures were carried out following the Mexican Official Standard (NOM-062-ZOO-1999). The experimental protocol was approved by the Bioethics Committee of the Faculty of Natural Sciences of the Universidad Autónoma de Querétaro (Querétaro, México) (5FCN2014).

After one week of acclimatization, the rats were randomized into four groups of eight rats each: (1) a standard diet group fed commercial rat food (48.7% carbohydrates (0.3% fructose), 5% lipids (1.6% saturated fat), and 23.9% proteins) and water; (2) an HFFD group fed a commercial standard diet (61.94%) with 18% food-grade fructose, 20% commercial pig lard, 0.03% vitamins, and 0.03% minerals (48.2% carbohydrates (18.2% fructose), 23.1% lipids (8.8% saturated fat), and 14.8% proteins) and water; (3) an HFFD + white group fed an HFFD supplemented with Alma Blanca HSC aqueous extract ad libitum; and (4) an HFFD + purple group fed an HFFD supplemented with Cuarenteña HSC aqueous extract ad libitum. Food and water were administered ad libitum; animals were weighed weekly, and food consumption was registered. The feed efficiency was calculated by dividing the food consumption by the average weight gain of the animals every week, and this value was divided by the seven days of the week and reported as feed efficiency/day Polyphenol, flavonoid, and anthocyanin consumptions were quantified for the different diets, according to the methods in Section 2.3, and the consumptions was estimated from the daily food intake. 

After sixteen weeks, the animals were sacrificed by guillotine decapitation after overnight fasting. The blood was collected and centrifugated to obtain serum and stored at −80 °C until use. Mesenteric adipose tissue was excised separately and weighed. The adipose tissue relative weight was calculated using the mesenteric total adipose tissue weight and the total body weight. Mesenteric adipose tissue was used for the histological analysis and was fixed in 10% neutral buffered formaldehyde at pH 7.4. Aliquots of serum samples were used to determine the fasting glucose (FG), fasting insulin (FINS), free fatty acids (FFA), and targeted metabolomics.

### 2.7. Histopathology of Mesenteric Adipose Tissue

Mesenteric adipose tissues were sectioned (3 μm) and stained with hematoxylin and eosin. Ten representative images were taken from each tissue (*n* = 8) with an optical microscope (Leica DM500). The adipocyte area was determined using ImageJ microscope software to estimate adipocyte hypertrophy [23].

### 2.8. Serum Fasting Glucose (FG), Fasting Insulin (FINS), and Free Fatty Acids (FFA)

Serum aliquots were used to determine the serum fasting glucose (FG) and insulin (FINS). FG was determined by the glucose oxidase method using a commercial colorimetric/enzymatic kit (Spinreact, Spain) and 10 µL of serum. FINS was measured using specific antibodies with a rat/mouse ELISA kit (EZRMI-13K^®^, Millipore, Burlington, MA, USA) and 10 µL of serum. Then, the homeostatic model assessment (HOMA) index was calculated to estimate the insulin resistance state (HOMA-IR). It was calculated as follows: fasting insulin (μIU/mL) × fasting glucose (mmol/mL)/22.5. The serum FFA concentration was measured following the method of the colorimetric microdetermination of nonesterified fatty acids in plasma reported by Duncombe [24], with 25 µL of serum and using palmitic acid as a standard.

### 2.9. Serum Targeted Metabolomics

Serum samples (200 µL) were homogenized with 400 µL of methanol. Samples were incubated for 10 min at −20 °C. Then, samples were centrifuged at 12,600× *g* for 15 min at 4 °C. Supernatants were recovered, filtered (Waters Co, Minispike Syringe Filter, PVDF, 13 mm, 0.2 µm), and placed in chromatographic amber vials for further analysis. The metabolomic profile was carried out in six biological replicates. Quality controls (QCs) were prepared by pooling 10 μL of each serum sample, whereas methanol was used as an analytical blank. QCs and blanks were injected prior the analytical method for system conditioning and every eight samples to ensure the stability of the analytical conditions. 

Samples (2 μL) were injected into an Acquity UPLC BEH C18 column (2.1 × 100 mm, 1.7 μm) at 35 °C in a UPLC-ESI-QToF MS system. The mobile phase consisted of solvents A (water with 0.1% formic acid) and B (acetonitrile with 0.1% formic acid) at a 0.4 mL/min flow rate. The solvent gradient was 95% A for 2 min, with a linear gradient of 95% to 5% A for 20 min and maintenance in the isocratic condition for 2 min. Then, a linear gradient of 5 to 95% A for 3 min was maintained for 2 min. A high-definition MS^E^ ionization was used in the negative ionization mode (ESI-). The mass range was established between 50 and 1800 Da. The mass spectrometer settings were set as previously described (Section 2.3). Raw data were acquired and processed using the UNIFI software (Scientific Information System, Waters Co.). A targeted analysis was carried out with the mass parameters described in Appendix A. Then, data were exported for statistical analysis by MetaboAnalyst 4.0.

### 2.10. Statistical Analysis

The results were expressed as means ± standard deviations. Differences among groups were determined by comparisons using the Tukey test (*p* < 0.05) with the JMP software (v11.0, SAS Institute). Metabolomics data were normalized by total ion abundance, cube-root-transformed, and Pareto-scaled. A multivariate statistical analysis was carried out through a principal component analysis (PCA) and a partial least-squares discriminant analysis (PLS-DA) on MetaboAnalyst 4.0 (https://www.metaboanalyst.ca (accessed on 7 November 2020)). Variable importance in projection (VIP) scores ≥ 1 were used to identify metabolites with discriminate power between the experimental groups. Then, discriminant metabolites were subjected to a univariate statistical analysis of their normalized intensities by the Kruskal–Wallis rank sum test (JMP software v11.0, SAS Institute, Cary, NC, USA).

## 3. Results

### 3.1. Phytochemical Characterization of Alma Blanca and Cuarenteña HSC Aqueous Extracts 

The monomeric anthocyanins were quantified in Alma blanca (identified as white), and Cuarenteña (identified as purple) HSC aqueous extracts; these compounds were only detected in the purple extracts (Table 1). Regarding the flavonoid content, no significant differences were found among the extracts (Table 1). Finally, the total phenolic compounds were up to 21% higher in the white extracts compared to the purple extracts (Table 1). 

The analysis of white and purple HSC aqueous extracts by *ESI-QToF-MS* revealed the presence of three anthocyanins, two flavones, twelve flavonols, five hydroxybenzoic acids, fourteen hydroxycinnamic acids, and six organic acids. Major differences were observed in the anthocyanin and organic acid compositions. As expected, anthocyanins were only identified in the purple extracts, with delphinidin sambubioside being the major pigmented compound. Conversely, white HSC aqueous extracts showed greater contents of organic acids compared to purple HSC aqueous extracts. On the other hand, similar overall compositions were observed for flavonoids and phenolic acids, with chlorogenic acid being the major phenolic acid of both HSC varieties.

### 3.2. Antioxidant Capacities of Alma Blanca and Cuarenteña HSC Aqueous Extracts

The antioxidant capacities of the aqueous extracts were determined by ABTS and DPPH assays. For the DPPH method, the values were 267 ± 8 and 298 ± 9 μmol Trolox equivalent/100 mL for Alma Blanca and Cuarenteña, respectively. These values were significantly different (*p* < 0.05). Regarding the DPPH method, the values were 183 ± 4 and 176 ± 11 μmol Trolox equivalent/100 mL, respectively. No significant differences were detected between the varieties (data not shown). The antioxidant values were similar to those reported by herbal beverages prepared as infusions, such as black tea and red tea, and higher than chamomile, which is one of the most consumed beverages worldwide [25].

### 3.3. Effect of Alma Blanca and Cuarenteña HSC Aqueous Extracts on Obesity and Its Complications in HFFD-Fed Rats

The body weights of animals fed an HFFD showed an increase of 38% compared to the standard diet group. Regarding the treatment groups, the consumption of white and purple HSC aqueous extracts decreased body weights by 11% and 9%, respectively, compared to the HFFD group (Figure 1A). 

Food intake was measured, and the results showed that the HFFD group consumed 18% less food per gram per day than the standard diet group (Figure 1B); however, they showed a higher feed efficiency (75%; Figure 1C) than the standard diet group. Rats supplemented with HSC aqueous extracts showed similar daily food intake and feed efficiency compared to the HFFD group.

Regarding the polyphenolic compound intake, rats supplemented with white and purple HSC aqueous extracts consumed 45 and 54 mL per day, respectively. No significant differences were observed in the daily intakes of total polyphenols (Figure 1D) and flavonoids (Figure 1C) from the white and purple HSC aqueous extracts, whereas a daily intake of 0.18 mg C3GE/kg/day was recorded for monomeric anthocyanins in purple HSC aqueous extract supplemented rats (Figure 1D).

As expected, the HFFD group showed enlarged adipocytes, as observed in the histology analysis (Figure 2A vs. Figure 2C), leading to an increased (*p* < 0.05) relative weight of the mesenteric adipose tissue (284.3%; Figure 2E) and an increased adipocyte area (128.2%; Figure 2E) compared to the standard diet group. The consumption of white and purple HSC aqueous extracts significantly (*p* < 0.05) reduced the relative weight of the mesenteric adipose tissue by 32.5% and 27.0%, respectively, compared to the HFFD group. Furthermore, the relative adipocytes area decreased by 16.3% and 13.5%. Accordingly, the histology analysis showed that adipocytes were slightly less enlarged compared to the HFFD group (Figure 2C,D vs. Figure 2B).

### 3.4. Insulin Resistance 

The administration of an HFFD for sixteen weeks did not significantly affect serum FG levels compared to the standard diet group. However, FINS, HOMA-IR, and FFA were increased (21.7, 55.8, and 47.7%, respectively; Table 2). The supplementation with white HSC aqueous extract significantly (*p* < 0.05) decreased FINS and HOMA-IR (33.4 and 35.8%, respectively) compared to the HFFD group, whereas no significant differences were observed with purple HSC aqueous extract supplementation. Moreover, white HSC aqueous extract decreased serum FFA by 22.0% compared to the HFFD group.

### 3.5. Serum Targeted Metabolomics

The targeted metabolomic profile allowed the identification of forty-five phospholipid species belonging to the following classes: five lysophosphatidic acids (LysoPA), eight lysophosphatidylethanolamines (LysoPE), three lysophosphatidylcholines (LysoPC), three lysophosphatidylinositols (LysoPI), six lysophosphatidylserines (LysoPS), two phosphatidic acids (PA), one phosphatidylcholine (PC), six phosphatidylethanolamines (PE), five phosphatidylinositol (PI), and six phosphatidylserines (PS).

The PCA score plot shows that rats were clustered into two distinct groups: (i) standard diet and (ii) HFFD; within this latter group, three subclusters were identified: HFFD, HFFD + white HSC aqueous extract, and HFFD + purple HSC aqueous extract (Figure 3A). The plot shows that the model described 46.7% of the total variance in the data set and that the variation between the different biological groups is more pronounced on PC1 (30.0%) than PC2 (16.7%). The 95% confidence curves of the HFFD + white HSC aqueous extract and HFFD + purple HSC aqueous extract groups overlap with the curve of the HFFD group, indicating a similar serum phospholipid profile.

Therefore, a supervised multivariate analysis was carried out using a PLS-DA model, which showed a performance of R^2^ = 0.80487 and a prediction of Q2 = 0.5838. The score plot showed a greater separation of the standard diet and the HFFD clusters; however, the clusters of both HSC aqueous extracts remained overlapped with the untreated HFFD group (Figure 3B). A quantitative estimation of the discriminatory power of the phospholipid species was obtained using the VIP score. Figure 4 shows the VIP score plot of the 30 phospholipid species with higher discriminant power. 

A univariate analysis was performed to evaluate the significance of the 16 best VIP scores (VIP ≥ 1), which are represented by boxplots (Figure 5) with normalized intensities. Ten species of phospholipids (LysoPC 16:1, LysoPC 18:0, PC 16:1 16:1, LysoPE 18:2, PE 16:0 22:4, LysoPS 18:2, LysoPS 20:5, PS 18:0 18:1, LysoPI 16:0, and LysoPI 20:4) decreased with HFFD, three phospholipids species (LysoPE 18:0, LysoPE 20:4, and PE 18:1 18:2) increased with HFFD, and the other three phospholipids did not show significant differences (data not shown). The consumption of white and purple HSC aqueous extracts increased the serum PC 16:1 level. Furthermore, the aqueous extracts of white roselle decreased LysoPE (18:0) and LysoPE (20:4), and the aqueous extracts of purple roselle increased LysoPI (16:0) and LysoPI (20:4) compared to the HFFD (Figure 5).

## 4. Discussion

In this study, we hypothesized that aqueous extracts elaborated from contrasting colored calyces of *H. sabdariffa* exert beneficial health effects through different mechanisms. The phytochemical characterizations demonstrated that white (Alma Blanca) and purple (Cuarenteña) HSC aqueous extracts have different compositions. Since anthocyanins are responsible for the HSC red/purple color, it was expected that these compounds would not be detected in white calyx extracts. The major anthocyanin identified in Cuarenteña HSC aqueous extract was delphinidin sambubioside, which along with cyanidin-3-*O*-glucoside has been reported as a major compound of the red varieties of HSC [12,13]. However, in this study cyanidin hexoside was detected at low levels. It is known that the concentration of roselle anthocyanins is affected by several factors, such as the cultivar and the climatic and geographical conditions [26]. Additionally, some processing factors must be considered during the elaboration of the aqueous extracts, such as light, high temperature, low acidity, and a pH above 3, which may affect anthocyanin stability and thus its final concentration [27].

On the other hand, no significant differences were observed in the total flavonoid contents between white and purple HSC aqueous extracts. Interestingly, several nonpigmented flavonoids identified in this study had previously been reported by other authors in different HSC varieties, such as myricetin hexoside, quercetin rutinoside (rutin), quercetin hexoside, quercetin, and myricetin, principally detected in purple extracts, and kaempferol pentosyl-hexoside in white extracts [11,14,28]. Conversely, to the best of our knowledge, this is the first study that identified the following flavonoids in HSC extracts: chrysoeriol hexoside, chrysoeriol apiosyl-hexoside, (iso)-rhamnetin rutinoside, kaempferol dihexoside, and quercetin hexoside-rhamnoside.

Regarding phenolic acids, 5-caffeoylquinic acid (chlorogenic acid) was the major phenolic acid that was found, followed by caffeoylquinic acid isomer II, and they were detected in similar concentrations in both HSC varieties. The presence of chlorogenic acid and its derivatives in HSC extracts has been reported by several studies, which is consistent with our findings [8,11,29]. The major organic acid found in white HSC aqueous extract was citric acid, whereas hibiscus acid was the major organic acid for purple HSC aqueous extract. Furthermore, to the best of our knowledge, this is the first study to report aconitic acid in HSC extracts.

White or purple HSC aqueous extracts were administered to HFFD-fed rats to evaluate their effect on obesity and its complications. Interestingly, despite the differential phytochemical compositions of these two varieties (white with higher contents of organic compounds and phenolic acids and purple with higher concentrations of flavonols and anthocyanins), the capacity they presented to prevent weight gain (9–11%) was similar in both varieties. The antiobesogenic effect of white HSC extract agreed with the results reported by Morales-Luna et al. [15]; these authors demonstrated that white and red extract varieties prevented an increase in body weight (8.9–11.2%) compared to rats fed an HFFD. Nevertheless, it is noteworthy, that most HSC studies have been carried out with red varieties, which are rich in anthocyanins similar to purple varieties, and have demonstrated that red varieties decrease body weight and adipocyte diameter [30].

Interestingly, the antiobesogenic effect of HSC aqueous extracts observed in this study was not associated with a decreased food intake or an altered feed efficiency. Similarly, Moyano et al. [30] and Morales-Luna et al. [15] reported that HSC extracts had no effect on appetite loss. In this study, HSC aqueous extracts provided daily intakes of 47–54 and 1.6–1.8 mg/kg/day of total polyphenols and flavonoids, respectively. Additionally, purple HSC aqueous extracts provided anthocyanins. This daily polyphenol intake was lower than that reported in similar studies with different sources. For instance, the polyphenol intake of the HSC aqueous extracts was 33–53% lower than that reported by Liu et al. [31] with green tea, whereas the flavonoid intake was up to 97% lower. Conversely, the anthocyanin intake of purple HSC aqueous extract in our study was 2.4 times higher than that reported by Morales-Luna et al. [15], who reported the antiobesogenic effect of a red HSC variety extract.

Excess energy ingested by animals fed an HFFD is stored as triglycerides in adipose tissue. Therefore, the beneficial effect of the consumption of white and purple HSC aqueous extracts on the mesenteric adipose tissue of HFFD-induced obese rats was evaluated. Interestingly, a similar trend was observed between weight gain, mesenteric adipose tissue relative weight, and adipocyte area, demonstrating that the antiobesogenic effects of both HSC aqueous extracts are related to decreased fat accumulation in adipose tissue.

Similar results have been reported previously. For instance, Villalpando-Arteaga et al. [32] reported that red HSC aqueous extracts reduced body weight and fat tissue accumulation in obese C57BL/6NHsd mice fed a high-fat diet, which was associated with their anthocyanin content. Moreover, Morales-Luna et al. [15] showed that white HSC aqueous extracts exerted an antiobesogenic effect, probably due to their high contents of organic acids, such as hibiscus acid and hydroxycitric acid, and flavonoids, such as quercetin and myricetin, which are associated with the prevention of body weight gain and adipocyte hypertrophy [12,33].

Our results suggest that the consumption of white HSC aqueous extract has a preventive effect on the development of obesity-induced insulin resistance, which may be related to decreased circulating FFA, which are well known to disrupt the insulin signaling cascade. To the best of our knowledge, this is the first study to report the effect of a white HSC aqueous extract on this metabolic alteration. It is noteworthy that the effects of the organic acids identified in white HSC aqueous extracts on insulin resistance have not been reported; therefore, further studies must be conducted to understand their metabolic implications.

Conversely, several authors have reported that anthocyanin-rich HSC varieties reduce serum FFA and ameliorate insulin resistance [5]; these beneficial health effects can also be associated with their high contents of nonpigmented flavonoids such as quercetin, myricetin, and kaempferol [34,35]. Interestingly, despite the high contents of these nonpigmented flavonoids and anthocyanins, no beneficial effect was observed in this study with the supplementation of purple HSC aqueous extracts on the development of insulin resistance.

The modulation of the serum phospholipid profile in obesity is related to the development of insulin resistance or dysregulated lipid metabolism. Moreover, it is known that the consumption of extracts rich in phenolic compounds, epigallocatechin gallate, quercetin, and gallic acid, among others, regulates these metabolic disorders [36]. Thus, we employed a targeted metabolomic–chemometrics combined approach to identify possible molecular targets of HSC aqueous extracts.

In this regard, the main phospholipids modified by the HFFD were those belonging to the ethanolamine and serine families, followed by the cholines and finally the inositol family. These results agree with previous studies that demonstrated decreases in the serum levels of different species of lysophosphatidylcholines in individuals with obesity or insulin resistance [37,38]. On the other hand, high HOMA-IR values had been positively related to some phospholipids such as PC C32:2 and LPC C14:0, while PC C43:6 and PC C44:12 were negatively associated with IR [39]. Additionally, other serum lysophospholipids have been associated with obesity, including LysoPS, LysoPI, and LysoPE; specifically, plasma LysoPE 18:2 is negatively correlated with obesity [40]. Both HSC aqueous extracts modified several serum phospholipids, mainly PC, lysoPE, and LysoPI species of 16 and 18 acyl chains. Similarly, the administration of polyphenol-rich berries to individuals with metabolic syndrome altered the serum phospholipid profile, mainly PE and PC [41]. Nevertheless, due to the limited current knowledge of the metabolic implications of these serum phospholipid alterations in the development of obesity and its complications, further studies are necessary to understand how these molecular targets can be related to the beneficial health effects observed with white and purple HSC aqueous extract supplementation in HFFD-fed rats.

## 5. Conclusions

White and purple roselle calyx aqueous extracts prevented the development of hypercaloric-diet-induced obesity, which was not related to satiety effects. Interestingly, differential effects were observed in the modulation of the serum phospholipid profile. Moreover, white roselle calyx aqueous extracts ameliorated the development of insulin resistance by decreasing circulating FFA; therefore, this latter variety exerted a greater beneficial health effect, which could be associated with its higher contents of phenolic and organic acids compared to the purple variety.

## Figures and Tables

**Figure 1 ijerph-19-16538-f001:**
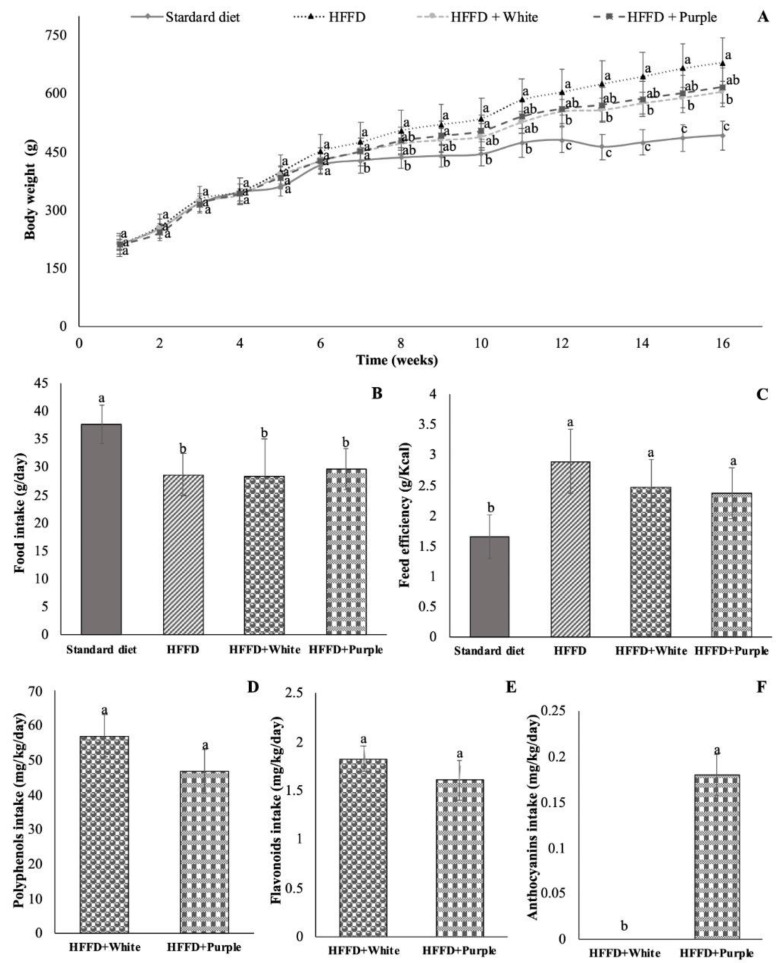
Body weight gain (**A**), food intake (**B**), feed efficiency (**C**), total polyphenol intake (**D**), flavonoid intake (**E**), and anthocyanin intake (**F**) of HFFD-fed rats supplemented with white and purple Hibiscus sabdariffa calyx aqueous extracts. Data are expressed as the mean values (*n* = 8), and bars indicate the standard deviations. Different letters indicate significant (*p* < 0.05) differences according to Tukey’s test. HFFD: high-fat and high-fructose diet.

**Figure 2 ijerph-19-16538-f002:**
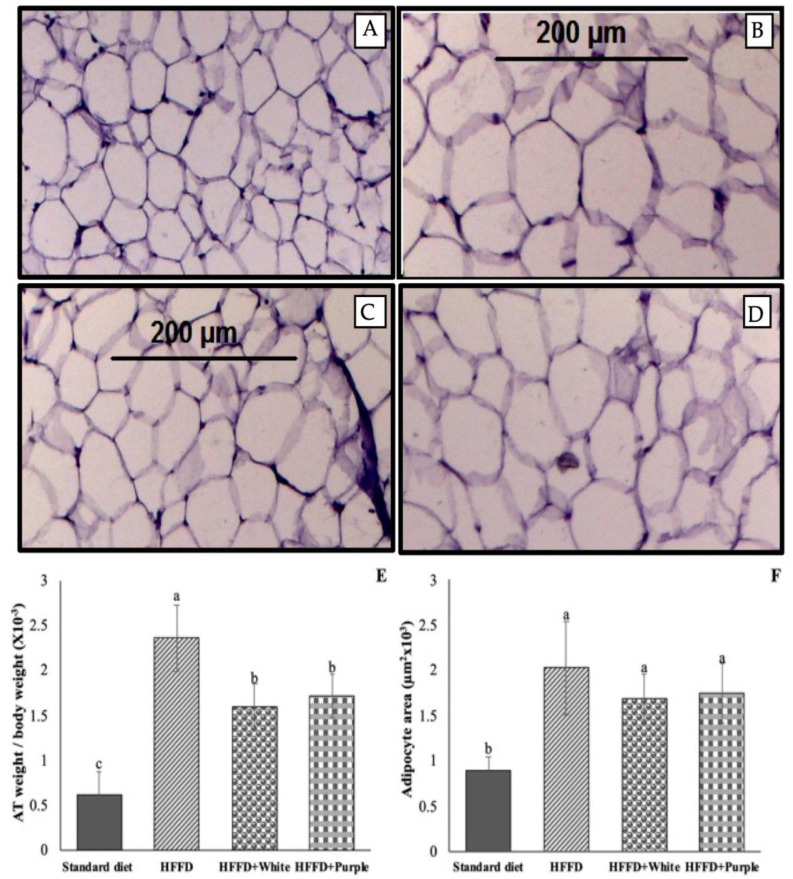
Mesenteric adipose tissue from HFFD-fed rats supplemented with white and purple Hibiscus sabdariffa calyx aqueous extracts. Histology analysis of (**A**) rats fed a standard diet, (**B**) rats fed an HFFD, (**C**) rats fed an HFFD supplemented with white HSC aqueous extract, and (**D**) rats fed an HFFD supplemented with purple HSC aqueous extract, (**E**) adipose tissue relative weight, and (**F**) adipocyte size. Magnitude of 10x for histology microphotography. Data are expressed as the mean values (*n* = 8), and bars indicate the standard deviations. Different letters indicate significant (*p* < 0.05) differences according to Tukey’s test. HFFD: high-fat and high-fructose diet; HSC: Hibiscus sabdariffa calyces; AT: adipose tissue.

**Figure 3 ijerph-19-16538-f003:**
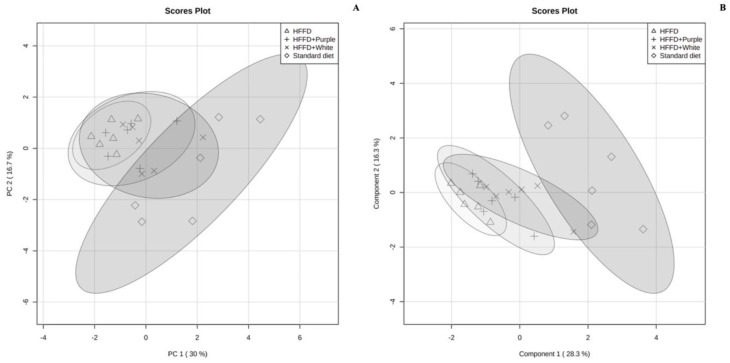
Principal component analysis (PCA) plot (**A**) and partial least-square-discriminant analysis (PLS-DA) plot (**B**) of the serum phospholipid profile of *HFFD-fed* rats *supplemented* with white and purple Hibiscus sabdariffa calyx aqueous extracts. The groups include rats fed a standard diet, rats fed an HFFD, rats fed an HFFD supplemented with white HSC aqueous extract, and rats fed an HFFD supplemented with purple HSC aqueous extract. HFFD: high-fat and high-fructose diet; HSC: *Hibiscus sabdariffa* calyces.

**Figure 4 ijerph-19-16538-f004:**
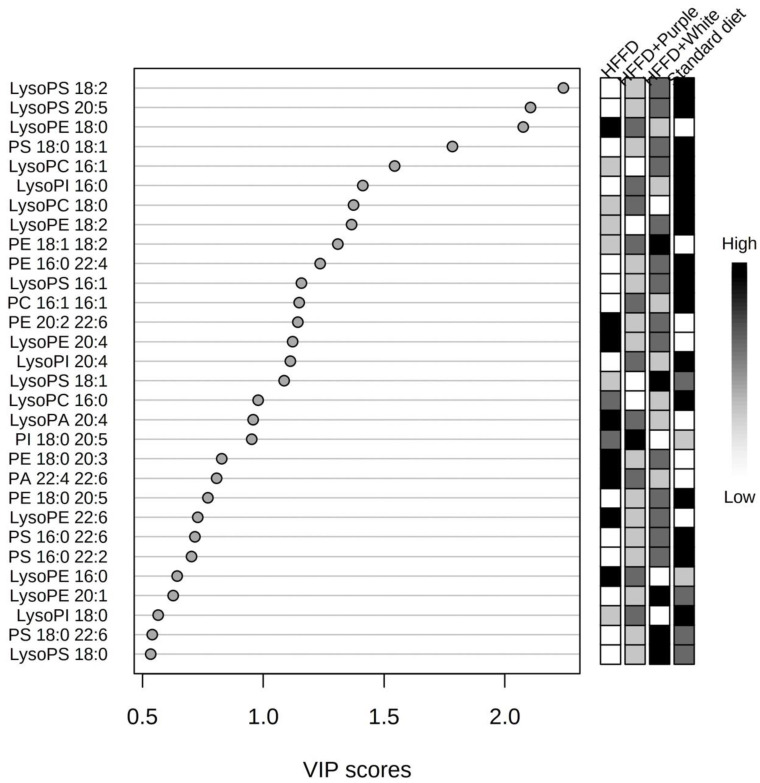
Variable importance in the projection (VIP) score plot obtained from partial least-square discriminant analysis (PLS-DA) of HFFD-fed rats supplemented with white and purple Hibiscus sabdariffa calyx aqueous extracts. HFFD: high-fat and high-fructose diet.

**Figure 5 ijerph-19-16538-f005:**
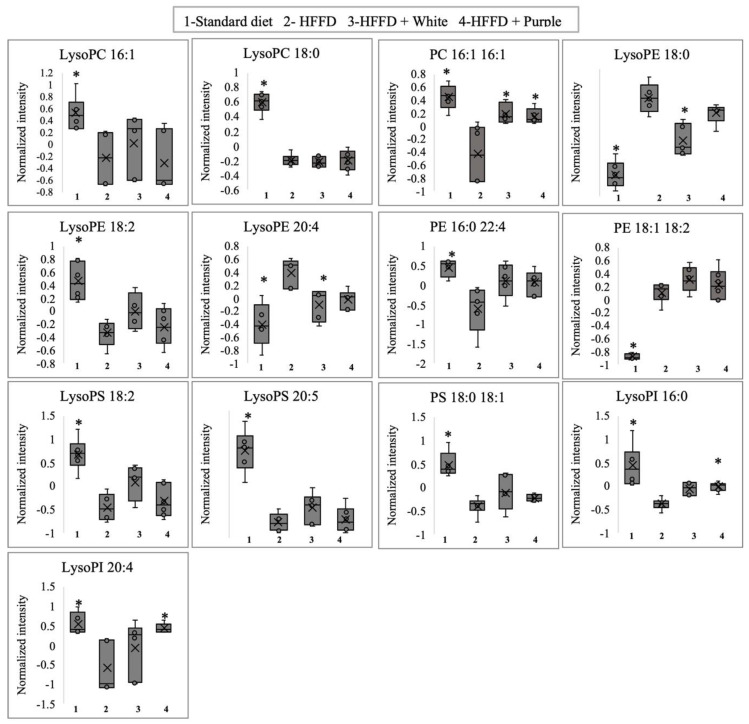
Discriminant serum phospholipids of HFFD-fed rats supplemented with white and purple Hibiscus sabdariffa calyx aqueous extracts. Data are expressed as the mean values (*n* = 6), and bars indicate the standard deviations. * Indicates a significant (*p* < 0.05) difference compared to the HFFD group according to the Kruskal–Wallis rank-sum tests. HFFD: high-fat and high-fructose diet.

**Table 1 ijerph-19-16538-t001:** Polyphenols and organic acid compositions of white and purple *Hibiscus sabdariffa* calyx aqueous extracts.

	Rt (min)	Molecular Formula	Observed Mass (Da)	Concentration	
				White	Purple
Monomeric anthocyanins ^1^	-------	-------	-------	ND	1.3 ± 0.08
Total flavonoids ^2^	-------	-------	-------	13.5 ± 0.72 a	11.93 ± 0.77 a
Total phenolic compounds ^3^	-------	-------	-------	417.0 ± 24.4 a	343.4 ± 24.4 b
**Anthocyanins ^4^**					
Delphinidin sambubioside **^+^**	2.25	C_26_H_29_O_16_	597.1455	ND	23.58 ± 0.84
Delphinidin hexoside **^+^**	2.70	C_21_H_21_O_12_	465.1070	ND	0.01 ± 0.00
Cyanidin hexoside **^+^**	2.95	C_21_H_21_O_11_	449.1117	ND	0.01 ± 0.01
**Flavones ^4^**					
Chrysoeriol hexoside	4.31	C_22_H_22_O_11_	462.1157	0.02 ± 0.00	0.05 ± 0.00
Chrysoeriol apiosyl-hexoside	4.85	C_27_H_30_O_15_	594.1577	0.82 ± 0.02	0.18 ± 0.01
**Flavonols ^4^**					
Kaempferol pentosyl-hexoside **^+^**	2.53	C_26_H_28_O_15_	580.1417	3.19 ± 0.12	0.03 ± 0.00
Myricetin hexoside **^+^**	3.41	C_21_H_20_O_13_	480.0903	0.63 ± 0.03	6.71 ± 0.81
Quercetin rutinoside (rutin) *,**^+^**	3.95	C_27_H_30_O_16_	610.1528	1.66 ± 0.10	9.02 ± 0.21
Quercetin hexoside **^+^**	4.18	C_21_H_20_O_12_	464.0960	1.69 ± 0.14	9.68 ± 1.39
(Iso)-rhamnetin rutinoside	4.31	C_22_H_22_O_11_	462.1157	0.08 ± 0.00	0.04 ± 0.00
Kaempferol dihexoside	4.74	C_27_H_30_O_16_	610.1514	0.10 ± 0.00	0.57 ± 0.20
Kaempferol hexoside-rhamnoside **^+^**	4.85	C_27_H_30_O_15_	594.1577	0.31 ± 0.01	1.44 ± 0.03
Quercetin hexoside-rhamnoside	5.04	C_27_H_30_O_16_	610.1511	0.06 ± 0.01	0.20 ± 0.01
Myricetin rhamnoside	5.09	C_21_H_20_O_12_	464.0954	0.09 ± 0.01	0.54 ± 0.10
Kaempferol hexoside **^+^**	5.14	C_21_H_20_O_11_	448.1011	0.16 ± 0.01	0.55 ± 0.06
Myricetin *,**^+^**	5.64	C_15_H_10_O_8_	318.0379	0.02 ± 0.00	2.70 ± 3.79
Quercetin *,**^+^**	8.74	C_15_H_10_O_7_	302.0421	0.04 ± 0.01	3.77 ± 6.49
**Hydroxybenzoic acids ^4^**					
Dihydroxybenzoic acid hexoside	1.82	C_13_H_16_O_9_	316.0787	14.52 ± 1.32	6.63 ± 0.14
Vanillic acid *,**^+^**	1.89	C_8_H_8_O_4_	168.0416	0.36 ± 0.03	0.14 ± 0.01
3,4-Dihydroxybenzoic acid (protocatechuic acid) *,**^+^**	1.90	C_7_H_6_O_4_	154.0261	0.10 ± 0.00	0.09 ± 0.00
Hydroxybenzoic acid (isomer I) **^+^**	2.39	C_7_H_6_O_3_	138.0313	0.17 ± 0.01	0.15 ± 0.01
Hydroxybenzoic acid (isomer II) **^+^**	5.12	C_7_H_6_O_3_	138.0316	0.45 ± 0.00	0.15 ± 0.01
**Hydroxycinnamic acids ^4^**					
5-caffeoylquinic acid (chlorogenic acid) *,**^+^**	2.15	C_16_H_18_O_9_	354.0948	33.88 ± 0.36	27.73 ± 0.99
Coumaroylquinic acid (isomer I) **^+^**	2.50	C_16_H_18_O_8_	338.1001	1.02 ± 0.09	1.32 ± 0.05
Caffeoylquinic acid (isomer II) **^+^**	2.69	C_16_H_18_O_9_	354.0956	28.04 ± 0.15	20.98 ± 1.00
Coumaric acid hexoside	2.71	C_15_H_18_O_8_	326.1004	0.02 ± 0.00	0.04 ± 0.00
Caffeic acid *,**^+^**	2.76	C_9_H_8_O_4_	180.0419	0.79 ± 0.24	1.27 ± 0.04
Coumaroylquinic acid (isomer II) **^+^**	3.09	C_16_H_18_O_8_	338.1007	1.37 ± 0.01	2.23 ± 0.09
Caffeoylquinic acid (isomer III) **^+^**	3.16	C_16_H_18_O_9_	354.0959	2.96 ± 0.82	3.88 ± 0.22
Feruloylquinic acid (isomer I) **^+^**	3.35	C_17_H_20_O_9_	368.1111	1.75 ± 0.06	0.31 ± 0.02
Coumaric acid *,**^+^**	3.47	C_9_H_8_O_3_	164.0466	ND	0.02 ± 0.00
Coumaroylquinic acid (isomer III) **^+^**	3.66	C_16_H_18_O_8_	338.0996	0.32 ± 0.00	0.26 ± 0.03
Feruloylquinic acid (isomer II) **^+^**	3.73	C_17_H_20_O_9_	368.1121	0.42 ± 0.02	ND
Ferulic acid *,**^+^**	3.90	C_10_H_10_O_4_	194.0578	0.02 ± 0.00	0.03 ± 0.00
Sinapic acid *,**^+^**	3.96	C_11_H_12_O_5_	224.0687	0.03 ± 0.00	0.05 ± 0.00
Dicaffeoylquinic acid (isomer I) **^+^**	4.98	C_25_H_24_O_12_	516.1281	0.07 ± 0.00	ND
**Organic acids ^4^**					
Quinic acid	0.92	C_7_H_12_O_6_	192.0264	0.042 ± 0.00	ND
Hydroxycitric acid **^+^**	2.14	C_6_H_8_O_8_	208.0217	0.07 ± 0.00	ND
Citric acid *,**^+^**	2.60	C_6_H_8_O_7_	192.0628	19.42 ± 0.11	1.74 ± 0.10
Hibiscus acid **^+^**	2.70	C_6_H_8_O_8_	190.0110	8.81 ± 1.12	6.37 ± 0.41
Benzoic acid	2.88	C_7_H_6_O_2_	122.0364	0.06 ± 0.00	ND
Aconitic acid	3.09	C_6_H_6_O_6_	174.0160	2.46 ± 0.27	1.99 ± 0.09

Data are expressed as the means of three independent determinations ± standard deviations. Different letters indicate significant differences according to the Tukey’s test (*p* < 0.05). ^1^ μg C3GE mL^−1^, ^2^ μg CE mL^−1^, ^3^ μg GAE mL^−1^, and ^4^ μg mL^−1^ of HSC aqueous extracts. * Identification confirmed with commercial standards. **^+^** Compounds reported previously in HSC extracts. ND: not detected.

**Table 2 ijerph-19-16538-t002:** Effects of white and purple *Hibiscus sabdariffa* calyx aqueous extracts on insulin resistance in high-fat and high-fructose diet fed rats.

Parameter	Standard Diet	HFFD	HFFD + White	HFFD + Purple
Serum FG ^1^	100.78 ± 18.2 a	129.36 ± 17.0 a	124.15 ± 15.2 a	116.44 ± 17.6 a
Serum FINS ^2^	1.54 ± 0.06 bc	1.88 ± 0.12 a	1.25 ± 0.10 c	1.83 ± 0.31 ab
HOMA-IR	9.53 ± 1.39 b	14.85 ± 1.08 a	9.52 ± 1.39 b	13.18 ± 3.25 ab
Serum FFA ^1^	18.74 ± 3.32 c	27.69 ± 1.21 a	21.59 ± 2.50 bc	24.88 ± 1.20 ab

Results are expressed as mean values (*n* = 8) ± standard deviations. Means within the same line with different letters indicate significant differences by Tukey’s test (*p* < 0.05). ^1^ mg dL^−1^, ^2^ ng mL^−1^. FG—fasting glucose; FINS—fasting insulin; FFA—free fatty acids; HFFD—high-fat and high-fructose diet.

## Data Availability

Not applicable.

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
