# Peer review of "Serum Phospholipids Are Potential Therapeutic Targets of Aqueous Extracts of Roselle (Hibiscus sabdariffa) against Obesity and Insulin Resistance"

_ijerph, 2022, doi:10.3390/ijerph192416538_

Round 1
Reviewer 1 Report
This study examined the effect of aqueous extracts of two varieties of Hibiscus sabdariffa Alma blanca (white-yellow color) and Cuarenteña (purple color) on the prevention of obesity and insulin resistance in rats fed a high-fat and fructose diet. The study design is appropriate and the research question is well-addressed. However, some minor issues need to be addressed before consideration for publication.
Comments:
1. The overall language needs to be improved. Several grammar issues are present. The authors will need to recheck the whole manuscript for presentation.
2. Besides the color and pigments content, are there any other differences between the two varieties, such as cultivar, growing environment, and chemical/molecular structure of the extracts? Any previous studies related to them?
3. The authors did measure the total phenolic contents, but how about the total antioxidant capacity? This is also another important parameter to look at when studying the beneficial effects of these bioactive compounds.
4. For the animal experiment part, what is the concentration of the added HSC extract? Is there any ratio applied? Need to specify.
5. Table 1 looks very dense. Much information seems not important at all. Those are not of interested. The authors could only keep molecular formulas and concentrations.
6. Some real-world implications could be discussed at the end. Are these extract safe to be supplemented into food products? Any studies on doses or side effects? In addition, some phenolics are sensitive to heat processing, will it be a problem in potential application?
Author Response
We appreciate your time and thank you for the observations.

Reviewer 2 Report
The main question of this research is whether Serum Phospholipids Are The Potential Therapeutic Targets Of Aqueous Extracts Of Roselle (Hibiscus Sabdariffa) Against Obesity And Insulin Resistance?
The antiobesity properties of roselle has been reported by a few researchers. However, the mechanism remains unclear.
Janson et al. (2021) has reported that Hibiscus sabdariffa L. calyx extract prevents the adipogenesis of 3T3-L1 adipocytes, and obesity-related insulin resistance in high-fat diet-induced obese rats.
This study shows that both roselle aqueous extracts prevented body weight gain, and white roselle extract ameliorated insulin resistance and decreased serum free fatty acids levels. Moreover, white roselle extract decreased 18:0 and 20:4 lysophosphatidylethanolamines and purple roselle extract increased 16:0 and 20:4 lysophosphatidylinositol as compared to HFFD-fed rats.
The conclusions consistent with the evidence and arguments presented and address the main question posed.
The references, tables, and figures are appropriate.
Author Response
We appreciate your time and your observations.
Reviewer 3 Report
My comments to the authors are enclosed.

Author Response

(The authors gave the same response as above.)
